# Selective Change in the Bacteria Cultured and Isolated in Respiratory Sputum from Elderly Patients during the SARS-CoV-2 Pandemic

**Masayuki Nagasawa** [1,2,*] [ID], **Tomoyuki Kato** [2,3], **Ippei Tanaka** [2,3] **and Emi Ono** [2,4]

1 Department of Pediatrics, Musashino Red Cross Hospital, Tokyo 180-8610, Japan
2 Department of Infection Control, Musashino Red Cross Hospital, Tokyo 180-8610, Japan; t.kato@musashino.jrc.or.jp (T.K.); i.b.d.a.p.9025@gmail.com (I.T.); e.ono@musashino.jrc.or.jp (E.O.)
3 Department of Pharmacy, Musashino Red Cross Hospital, Tokyo 180-8610, Japan
4 Department of Laboratory, Musashino Red Cross Hospital, Tokyo 180-8610, Japan
* Correspondence: masayukin@musashino.jrc.or.jp

**Abstract:** The SARS-CoV-2 pandemic has affected social patterns and consequently the prevalence of infections, such as seasonal influenza. It has been reported that invasive pneumococcal infection has markedly decreased worldwide. Method: We retrospectively investigated the bacteria cultured and isolated from 23,052 respiratory sputum samples obtained at our hospital from April 2015 to March 2022. The average patient age was 71.8 years old, with a standard deviation of 16.0 years old. There was no significant difference in the age of the patients or the female-to-male ratio between each year. The detection ratio of bacteria was analyzed in accordance with sputum quality based on the Geckler classification. Results: The detection ratio of community-acquired pneumonia pathogens such as *Haemophilus influenzae*, *Moraxella catarrhalis*, and *Streptococcus pneumoniae* increased in parallel with the quality of the sputum, while that of hospital-acquired pneumonia pathogens such as *Klebsiella pneumoniae*, *Pseudomonas aeruginosa*, and *Staphylococcus aureus* was not significantly affected by the quality of the sputum. The detection ratio of former pathogens in the good-quality respiratory sputum had decreased significantly since April 2020 by 60–80%, while that of *P. aeruginosa* and *S. aureus* had increased by 40–50%. Conclusions: The SARS-CoV-2 pandemic reduced the detection ratio of *H. influenzae*, *M. catarrhalis*, and *S. pneumoniae* but increased that of *P. aeruginosa* and *S. aureus* in the good-quality respiratory sputum from elderly patients. The influence of this selective change in isolated bacteria on the health and comorbidity of elderly patients remains to be investigated.

**Keywords:** SARS-CoV-2 pandemic; Haemophilus influenzae; Streptococcus pneumoniae; community-acquired pneumonia; hospital-acquired pneumonia; respiratory sputum; Geckler classification

## 1. Introduction

It is well known that the pandemic caused by the new coronavirus, systemic acute respiratory syndrome coronavirus 2 (SARS-CoV-2), is not only a global health problem but also has had a great impact on the global economy and changes in people's social behavior [1,2]. Infectious diseases are closely related to people's social activities. The SARS-CoV-2 pandemic has markedly reduced the worldwide seasonal influenza prevalence [3–6] and invasive pneumococcal infections [7–10]. On the other hand, it has been reported that the prevalence of rhinovirus/enterovirus (RV/EV) infection was not affected during the SARS-CoV-2 pandemic [11,12]. One of the reasons is that RV/EV infections are perennial and predominantly prevalent among children, especially in infants, contrary to influenza, which is seasonal and to which all generations are susceptible during the flu season. It is difficult for children in this age group to follow standard infection control procedures and maintain social distancing in daily life. Virus transmission may occur in the family, between parents and young children, and may spread into adult society consequently, and

vice versa. Only social lockdowns are useful for the prevention of these types of virus infections among young children. In fact, the social lockdown enforced from April to May of 2020 throughout Japan, in which all of the nurseries were closed, restrained the seasonal endemic of respiratory syncytial virus, which is usually prevalent in young children from summer to late autumn every year. It resumed in the 2021 and 2022 seasons again in Japan.

The respiratory tract's bacterial flora (microbiome) may affect the immune response and the severity of respiratory tract viral infections, including SARS-CoV-2. In this sense, it is interesting and significant to investigate and compare the trends and changes in oral bacteria cultures and isolated before and after the SARS-CoV-2 pandemic [13].

It is well known that there are two types of pneumonia among elderly people; one is community-acquired pneumonia (CAP), in which *Streptococcus pneumoniae*, *Haemophilus influenzae*, and atypical bacteria such as *Mycoplasma pneumoniae* and *Chlamydophila pneumoniae* are the main causative pathogens. The other is hospital-acquired pneumonia (HAP), in which multidrug-resistant organisms (MDROs) such as *Pseudomonas aeruginosa*, *Acinetobacter baumannii*, *Klebsiella pneumoniae*, and methicillin-resistant *Staphylococcus aureus* (MRSA) are the main causative pathogens. In both cases, the etiology may also depend on underlying health conditions, immunocompromised status, and previous antibiotic therapy. Therefore, we focused especially on *H. influenzae*, *Moraxella catarrhalis*, and *S. pneumoniae* as pathogenic bacteria for CAP, and *K. pneumoniae*, *P. aeruginosa*, and *S. aureus* as pathogenic bacteria for HAP, in this analysis.

Considering this background, we investigated the effects of the SARS-CoV-2 pandemic on bacteria cultured and isolated from respiratory sputum, focusing on elderly patients (≥60 years old) by analyzing the samples obtained at our hospital in accordance with the quality of the sputum based on the Geckler classification between April 2015 and March 2022.

## 2. Method

The Musashino Red Cross Hospital is a tertiary emergency medical facility in the North Tama area of metropolitan Tokyo, adjacent to the west of Central Tokyo, Japan, which includes more than one million residents. It has 611 beds, more than 20,000 annual admissions, more than 10,000 annual emergency transfers, and approximately 1800 outpatients per day.

Respiratory sputum samples obtained from inpatients or outpatients to evaluate respiratory infection at our hospital between April 2015 and March 2022 were investigated. Data were obtained from the electrical database of the laboratory section, combined with each patient's information from the electronic medical record. The quality of each sample was graded based on the Geckler classification system [14], and the detection ratio of each bacterium in the graded samples was calculated and compared. The Geckler classification system is a microscopic method of evaluating the quality of sputum for microbiological testing. It is clinically considered that sputum graded as Geckler 1, 2, and 3 is not suitable for further bacterial examinations such as Gram staining and isolation culture, and sputum graded as Geckler 4 and 5 is suitable for isolation culture to detect pathogenic microorganisms (Supplementary Table S1). In response to a strong request from clinicians, Geckler 1, 2, and 3 graded sputum samples were also transferred for isolation culture at our hospital during the study period. The determination of grades was performed by a specialist bacterial laboratory technician. Individual sputum was Gram-stained and spread on a solid medium and the detected colonies were plated again on a selective medium for the identification of the bacterial species. On the other hand, antimicrobial susceptibility was determined by the disc plate method. The procedure used to culture and identify the species of bacteria was performed according to the Japanese Committee for Clinical Laboratory Standards [15], which complies with the Clinical and Laboratory Standards Institute (CLSI) [16].

Principally, the analysis was performed using the first detected bacterium. However, the same analysis was also performed using the sum of the first, second, and third detected bacteria (data not shown). Statistical analysis was performed by Student's *t*-test and the

chi-square test, and $p < 0.05$ was determined as significant. Statistical analyses were carried out using JMP 14 (SAS Institute, Cary, NC, USA).

This retrospective study was approved by the Clinical Research Ethical Committee of Musashino Red Cross Hospital as No. 3028.

## 3. Results

The total sample number analyzed was 23,052. The annual sample number and the average patient age are presented in Table 1, and there was no significant difference in the average patient age or female-to-male ratio between each year. The average age of all patients was 71.8 years old, with a standard deviation of 16.0 years old. The age distribution of the patients from whom the respiratory sputum samples were obtained is presented in Table 2. More than 90% of the patients were 50 years old or older. Figure 1 shows the distribution of the Geckler classification in the samples by each fiscal year. The ratio of Geckler 1, 2, and 3 had increased significantly since April 2020, which was considered partly due to the SARS-CoV-2 pandemic, because the splash-producing procedure, which is necessary to obtain good-quality sputum, was restricted to some extent to prevent the unexpected spread of SARS-CoV-2 from asymptomatic infected persons. On the other hand, the ratio of Geckler 5 decreased significantly as opposed to that of Geckler 1, 2, and 3. We analyzed not only the absolute number of detected bacteria but also the ratio of each bacterium detected in each Geckler class, to avoid the effect of annual variation in the ratio of Geckler classes. Figure 2 presents the detection ratio of each bacterium according to the Geckler classification. The detection ratio of *H. influenzae*, *M. catarrhalis*, and *S. pneumoniae* increased in parallel with the quality of the sputum, while that of *K. pneumoniae*, *P. aeruginosa*, and *S. aureus* was not significantly affected by the quality of the sputum. *Candida species* and α-*Streptococcus species*, both of which are indigenous microorganisms in the oral cavity, represented the majority of residual bacteria detected in each classified sputum sample. This trend was consistently and equally found between each year.

**Table 1.** The annual respiratory sputum sample numbers between April 2015 and March 2022.

| | 2015.4–2016.3 | 2016.4–2017.3 | 2017.4–2018.3 | 2018.4–2019.3 | 2019.4–2020.3 | 2020.4–2021.3 | 2021.4–2022.3 |
|---|---|---|---|---|---|---|---|
| sample number | 2814 | 2960 | 3152 | 3498 | 3583 | 3456 | 3589 |
| average age(y) | 69.32 | 70.73 | 72.11 | 72.58 | 72.67 | 72.68 | 72.51 |
| median age (y) | 74 | 74 | 75 | 76 | 76 | 76 | 75 |
| SD (y) | 18.18 | 16.66 | 15.59 | 15.96 | 15.90 | 15.00 | 14.60 |
| female ratio | 0.37 | 0.39 | 0.36 | 0.42 | 0.43 | 0.37 | 0.39 |

The annual respiratory sputum sample number, average age, median age, and female-to-male ratio are presented. SD: standard deviation.

**Table 2.** The respiratory sputum sample numbers by age.

| Age (y) | Sample Number | % |
|---|---|---|
| 0–14 | 95 | 0.4 |
| 15–19 | 104 | 0.5 |
| 20–29 | 382 | 1.7 |
| 30–39 | 618 | 2.7 |
| 40–49 | 1035 | 4.5 |
| 50–59 | 1831 | 7.9 |
| 60–69 | 3711 | 16.1 |
| 70–79 | 6987 | 30.3 |
| 80–89 | 6585 | 28.6 |
| 90–99 | 1669 | 7.2 |
| 100- | 35 | 0.2 |

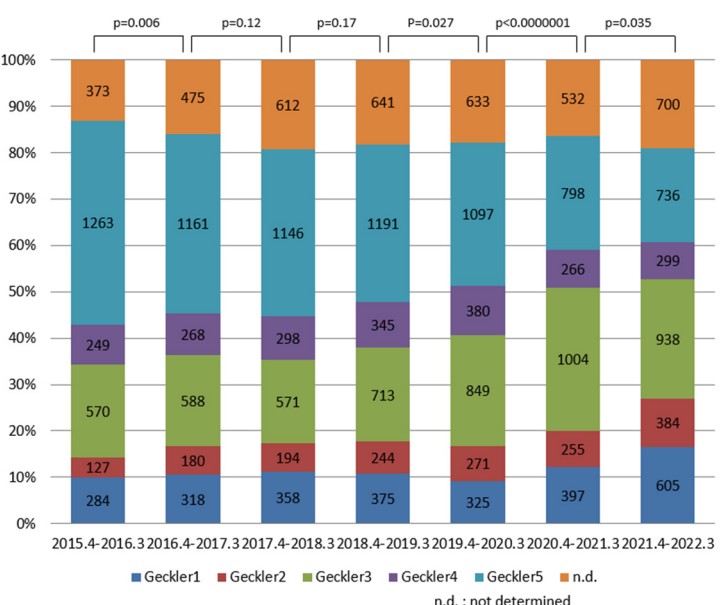

**Figure 1.** The annual distribution (%) of the respiratory sputum samples based on the Geckler classification. The ratio of Geckler 5 sputum was compared between each fiscal year by chi-square analysis. The number in the column represents each sample number considered for Geckler classification.

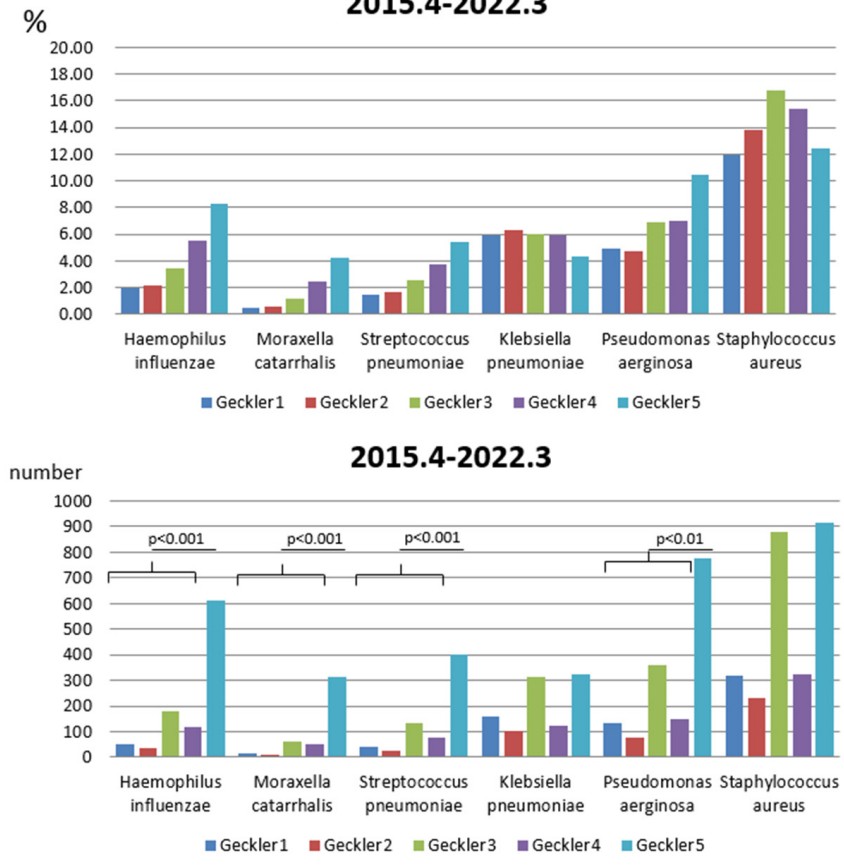

**Figure 2.** The detection ratio (%) of each bacterium from the respiratory sputum according to the Geckler classification is presented (**upper graph**). The sample number (number) of each detected bacterium from the respiratory sputum according to the Geckler classification is also presented (**lower graph**). In both graphs, the first detected bacterium was counted. Numerical data are presented in Supplementary Table S2. Data of Geckler 5 were compared to those of Geckler 1–4 by Student's *t*-test.

The respiratory sample number by age and its ratio is presented. More than 90% of samples were from patients aged 50 years old or older.

The detection ratio of each bacterium in the samples classified as Geckler 5 in each year is presented in Figure 3. Notably, the absolute detected number and the detection ratio of *H. influenzae*, *M. catarrhalis*, and *S. pneumoniae* significantly decreased by 60–80% ($p < 0.001$) after April 2020 compared to 2015–2019 for two years in a row. The result was the same when combining the data of Geckler 3, 4, and 5 or Geckler 4 and 5. However, the number of *K. pneumoniae*, *P. aeruginosa*, and *S. aureus* detected did not change significantly. On the other hand, the detection ratio of *P. aeruginosa* in 2021, and that of *S. aureus* in both 2020 and 2021, increased significantly. The results were the same when the detection ratio of the sum of the first, second, and third detected bacteria was analyzed. These tendencies were not observed in the Geckler 1 sputum (Supplementary Figure S1).

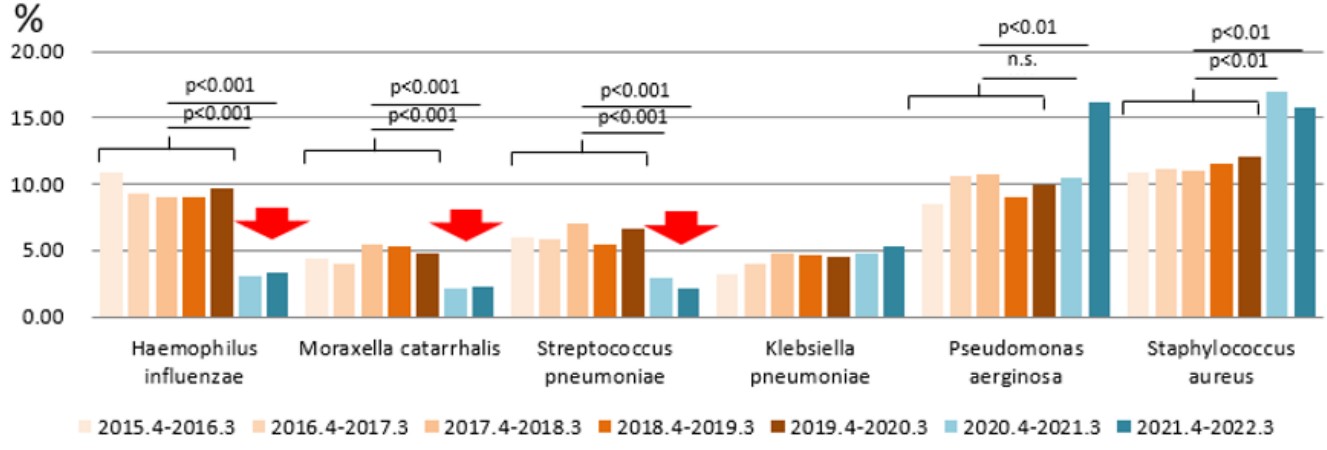

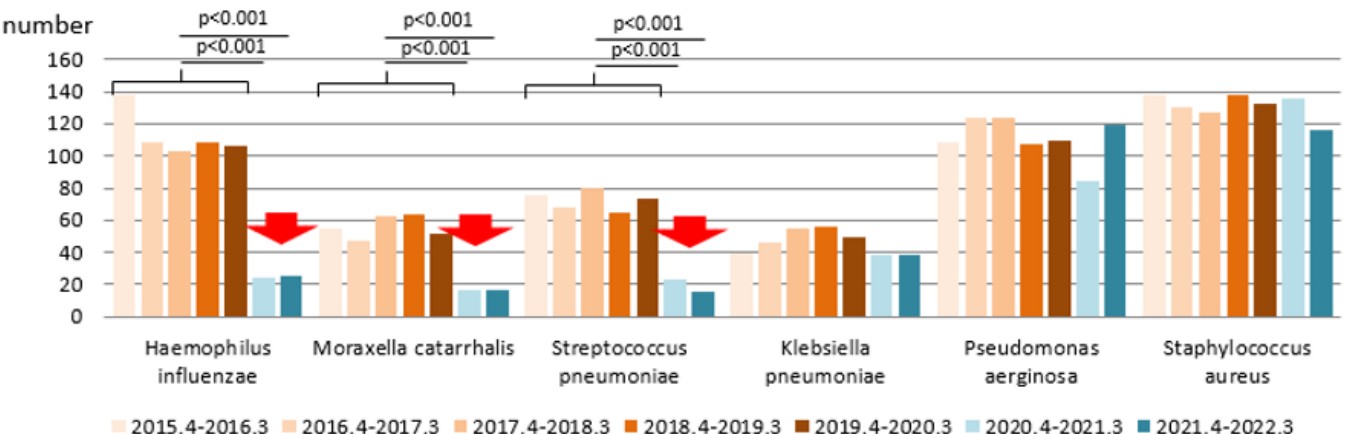

**Figure 3.** The trend of the detection ratio (%) of each bacterium from the respiratory sputum classified as Geckler 5 is presented (**upper graph**). The trend of the respiratory sputum sample number (number) classified as Geckler 5 in which each bacterium was detected is also presented (**lower graph**). In both graphs, the first detected bacterium was counted. The data of 2020.4–2021.3 and 2021.4–2022.3 were compared to those of 2015.4–2020.3 using Student's *t*-test. Numerical data are presented in Supplementary Table S3.

## 4. Discussion

*H. influenzae*, *M. catarrhalis*, and *S. pneumoniae* are the most frequently detected pathogens of bacterial pneumonia in children [17–19]. These bacteria are transmitted and spread between persons and induce respiratory infection. Additionally, in elderly people ($\geq$65 years old), these pathogens represent an important aspect of community-acquired pneumonia [20–22]. In contrast, *K. pneumoniae*, *P. aeruginosa*, and *S. aureus* are the main

causative pathogens of hospital-acquired pneumonia, such as aspiration pneumonia [23]. This is because elderly people may have compromised immune systems or other medical conditions that make them more susceptible to these types of infections.

In this study, it was clearly shown that the detection ratio of *H. influenzae*, *M. catarrhalis*, and *S. pneumoniae* increased in parallel with the quality of the sputum, while the detection ratio of *K. pneumoniae*, *P. aeruginosa*, and *S. aureus* was not significantly affected by the quality of the sputum. This finding indicates that these bacteria were possibly detected in sputum from elderly patients irrespective of pneumonia. In light of these observations, it is quite interesting that the detection ratio of *H. influenzae*, *M. catarrhalis*, and *S. pneumoniae* was reduced significantly during the SARS-CoV-2 pandemic after April 2020 for two consecutive years. *H. influenzae*, *M. catarrhalis*, and *S. pneumoniae* are considered to be transient and exogenous bacteria in nature, although they may behave as resident bacteria in some patients, especially in infants and young children. Since the onset of the SARS-CoV-2 pandemic, social activities and the flow of people have been restricted and reduced, and public hygiene procedures such as universal mask wearing and hand washing are encouraged. It has been reported that invasive infection with *H. influenzae*, *S. pneumoniae*, and *Neisseria meningitidis* has been reduced since the onset of the SARS-CoV-2 pandemic [8]. In this context, it is concluded that the transmission of *H. influenzae*, *M. catarrhalis*, and *S. pneumoniae* among elderly people was reduced during the SARS-CoV-2 pandemic. *K. pneumoniae*, *P. aeruginosa*, and *S. aureus* are considered mainly endogenous pathogens; therefore, the detection ratio of these bacteria did not change significantly during the SARS-CoV-2 pandemic. It is interesting that the detection ratio of *P. aeruginosa* was partly affected by the quality of the sputum, indicating transmission between patients and possibly healthcare workers in part. However, the detected number of *P. aeruginosa* was not affected significantly by the SARS-CoV-2 pandemic. This may indicate an increase in hospital-acquired pneumonia caused by *P. aeruginosa*.

In this regard, caution must be exercised in the interpretation of the results, because the quality of the sputum significantly deteriorated in 2020 and 2021 (Figure 1). Although the number of detected *P. aeruginosa* and *S. aureus* pathogens in Geckler 5 sputum did not change, the detection ratio of the two pathogens increased in 2020 and 2021. In November 2020, the highly toxic B.1.1214 strain of SARS-CoV-2 was prevalent as a third wave, and many elderly patients ($\geq$65 years old) required ventilation at our hospital. Vaccination for SARS-CoV-2 was started initially for medical staff in March 2021 and for elderly people in June 2021 in Japan. In July 2021, the moderately toxic and highly infectious δ variant strain of SARS-CoV-2 prevailed as a fifth wave and a number of middle-aged patients with SARS-CoV-2 infection required long-term ventilation and intensive care in our hospital. At the peak of the fifth wave, we experienced a shortage of ventilators, because the number of patients requiring long-term ventilation drastically increased. It has been reported that long-term ventilation is the most significant risk factor for HAP by *P. aeruginosa* and *S. aureus* [24]. It is considered that the increased detection ratio of *P. aeruginosa* might be attributed to the dramatic increase in the δ variant strain of SARS-CoV-2.

The transmission or spread of pathogenic bacteria that cause pneumonia in elderly people can occur through several routes.

The first is inhalation. The inhalation of airborne droplets containing pathogenic bacteria, such as *S. pneumoniae*, *H. influenzae*, or *Legionella pneumophila*, can lead to pneumonia. The second is aspiration. The aspiration of oral or gastric content can introduce bacteria, such as *K. pneumoniae* or *S. aureus*, into the lungs, leading to aspiration pneumonia. The third is contact. Direct contact with infected individuals or contaminated surfaces can also lead to the transmission of bacteria that cause pneumonia, such as methicillin-resistant *S. aureus* (MRSA) or multidrug-resistant *P. aeruginosa*.

Elderly people are particularly susceptible to pneumonia due to age-related changes in the respiratory system, such as a decreased cough reflex, weakened immune response, and underlying medical conditions, which can make them more vulnerable to infections. Additionally, elderly people living in long-term care facilities or hospitals are at a higher

risk of acquiring pneumonia due to close contact with other patients, shared living spaces, and frequent exposure to healthcare workers. In this context, elderly people were most vulnerable to the effects of changing social activities caused by the SARS-CoV-2 pandemic in many respects.

As previously mentioned, *H. influenzae* and *S. pneumoniae* are the major pathogens of respiratory infection during childhood, and it has been reported that they represent more than 80% of bacterial community-acquired pneumonia cases in patients under 15 years old and more than 95% in patients under two years old [17,18], which is a much higher ratio than that in the elderly patients observed in this study. The sample number from children (0–14 years old) represented as little as 0.4% of the total sample number in this study (Table 2). *H. influenzae* and *S. pneumoniae* were detected only in seven and eleven Geckler 5 graded sputum samples from children, respectively. This shows that our results almost entirely reflected bacteria cultured and isolated in respiratory sputum from elderly patients. Although the respiratory sputum sample number obtained from patients below 60 years old represented only 17.7%, the trend of the detected bacteria was almost similar to that from patients aged 60 years and older (Supplementary Figure S2). This indicates that this selective change in the bacteria cultured and isolated in the respiratory sputum was observed even in the middle-aged population. We have recently reported that respiratory viruses are transmitted between children and adults reciprocally in the community from epidemiological observations [25]. Although we did not have sufficient data on bacteria cultured and isolated from the respiratory sputum of children, it is feasible to speculate that *H. influenzae* and *S. pneumoniae* infections in children were also reduced during the SARS-CoV-2 pandemic, which also led to the reduction in the detection ratio of *H. influenzae* and *S. pneumoniae* in the respiratory sputum from the elderly patients in a reciprocal manner. In accordance with this, it has been reported that invasive pneumococcal infection has been markedly reduced in children as well as in elderly people worldwide [26,27]. The number of *S. pneumoniae* bacteremia cases decreased significantly, but that of *S. aureus* bacteremia did not change during the SARS-CoV-2 pandemic in our hospital. Furthermore, national surveillance in Japan shows a marked reduction in streptococcosis, which is caused by Group A *Streptococcus pyogenes*, in children for two years in a row since April 2020 [19].

There were several limitations in our study. The first is that we could not validate the time of sputum sampling or the clinical background of the patients, including comorbidities and antibiotic treatment. The bacteria in the sputum in elderly people may vary depending on the geographical location, local antibiotic resistance patterns, and individual patient factors. No significant changes were observed in the antibiogram of our hospital during the analysis period for the targeted bacteria. The second is that each sputum examination was performed at the discretion of each physician, and there were no clear criteria for the indication of sputum sampling in this cohort. The third is that this study was performed retrospectively. The fourth is that this study was conducted in a single institute and it might have been affected by facility characteristics. In this study, we could not evaluate the relation between selective changes in the bacteria cultured and isolated from the respiratory sputum and the severity of respiratory tract viral infection, including SARS-CoV-2. This issue seems to be interesting and is worth investigating in the future.

However, even considering these limitations, our long-term retrospective study clearly presents the different transmission characteristics of respiratory pathogenic bacteria among elderly patients and the impact of the SARS-CoV-2 pandemic on the bacteria cultured and isolated from the respiratory sputum in elderly patients. These changes had never been observed at our hospital before the onset of the SARS-CoV-2 pandemic. This also indicates the usefulness, effectiveness and limitations of infection control procedures performed in society in general and in medical facilities during the SARS-CoV-2 pandemic. The influence of this selective change observed on the health and comorbidities of elderly patients remains to be investigated in the future.

## 5. Conclusions

The detection ratio of *H. influenzae, M. catarrhalis*, and *S. pneumoniae* in the good-quality respiratory sputum from elderly patients markedly decreased, and that of *P. aeruginosa* and *S. aureus* increased significantly, after the onset of the SARS-CoV-2 pandemic. This might be attributed to the changes in social activities and comorbidities brought about by the SARS-CoV-2 pandemic. The influence of this selective change in bacteria cultured and isolated from respiratory sputum on the health and comorbidities of elderly patients remains to be investigated.

**Supplementary Materials:** The following supporting information can be downloaded at: https://www.mdpi.com/article/10.3390/applmicrobiol3030068/s1, Figure S1: The trend of the detection rate ratio (%) of each bacterium from the respiratory sputum classified as Geckler1 is presented (upper graph). The trend of the respiratory sputum sample number (number) classified as Geckler1 in which each bacterium was detected is presented (lower graph). In both graphs, the first detected bacterium was counted.; Figure S2: The trend of the detection rate ratio (%) of each bacterium from the respiratory sputum classified as Geckler5 is presented. Upper graph shows the data from the patients aged 60 years and older. Lower graph shows the data from the patients aged below 60 years. In both graphs, the first detected bacterium was counted. The data of 2020.4–2021.3 and 2021.4–2022.3 were compared to that of 2015.4–2020.3 by using Student's *t*-test; Table S1: The Geckler classification; Table S2: The detection ratio (%) of each bacterium from the respiratory sputum according to the Geckler classification is presented (upper table). The sample number of each detected bacterium from the respiratory sputum according to the Geckler classification is presented (lower table). In both tables, the first detected bacterium was counted. Right column presents average and standard deviation of Gecker1–3 (upper) and Geckler1–4(lower); Table S3: The trend of the detection ratio (%) of each bacterium from the respiratory sputum classified as Geckler5 is presented (upper table). The trend of the respiratory sputum sample number classified as Geckler5 in which each bacterium was detected is presented (lower table). In both tables, the first detected bacterium was counted. Right column presents average and standard deviation of 2015.4–2020.3 data

**Author Contributions:** M.N.: conceptualization, writing—original draft, review and editing, formal analysis. T.K.: data curation, writing—review. I.T.: data curation. E.O.: data curation. All authors have read and agreed to the published version of the manuscript.

**Funding:** This research received no external funding.

**Institutional Review Board Statement:** The study was approved by the Clinical Research Ethical Committee of Musashino Red Cross Hospital as No. 3028.

**Informed Consent Statement:** Informed consent was secured by the opt-out method.

**Data Availability Statement:** Data available on request from the authors.

**Conflicts of Interest:** The authors have no conflict of interest to declare.

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
