# Peer review of "Selective Change in the Bacteria Cultured and Isolated in Respiratory Sputum from Elderly Patients during the SARS-CoV-2 Pandemic"

_2673-8007, doi:10.3390/applmicrobiol3030068_

Round 1

Reviewer 1 Report (Previous Reviewer 2)

I revised the manuscript. I indicated corrections on the manuscript.

Manuscript must be revised by an English speaker.

Author Response

Reviewer 2 Report (Previous Reviewer 3)

The current work focuses on the Selective change in the bacteria cultured and isolated in respir-2 atory sputum from elderly patients during the SARS-CoV-2 3 pandemic. The experimental work appears to have been carried out well. However, a few points deserve attention for further publication. I suggest that it is accepted for publication after the following revisions:

 - Authors should make clear in the manuscript the effect of Bacterial Co-Infections: In addition to SARS-CoV-2 viral infection, elderly patients may develop secondary bacterial infections such as bacterial pneumonia. These infections can result in the isolation of different bacterial species in the respiratory sputum of patients. How was coinfection identified and treated to improve patient health?

 - About susceptibility to respiratory infections: The elderly, in general, are more susceptible to respiratory infections due to the aging of the immune system and other comorbidities. This may influence the bacterial composition present in respiratory sputum, as certain bacterial species may find a more favorable environment for colonization and growth. How was this identified in the manuscript?

 - Associated respiratory complications: Some elderly patients with COVID-19 may develop respiratory complications such as acute respiratory failure or severe pneumonia. These conditions can affect lung health and alter the bacterial microbiota of the respiratory tract. These changes may have implications for response to treatment and patient recovery. How was this accomplished in the manuscript?

 - Microbiological monitoring: During the SARS-CoV-2 pandemic, it is essential to monitor the respiratory microbiota of elderly patients to detect relevant bacterial changes. This can help clinicians make appropriate treatment decisions, including the appropriate use of antibiotics when needed, to prevent the development of bacterial resistance and ensure treatment effectiveness. This discussion must be included in the manuscript.

 - Extensive use of antibiotics: During the pandemic, some elderly patients with COVID-19 may be treated with antibiotics to prevent or treat secondary bacterial infections. The indiscriminate or inappropriate use of antibiotics can lead to selective alterations in the bacteria present in the respiratory tract. Some bacteria can acquire resistance to the antibiotics used, which can result in the increase of resistant bacterial strains. This discussion must be included in the manuscript.

 - Disruption of the respiratory microbiome: SARS-CoV-2 infection and the hospital environment can alter the respiratory microbiome of elderly patients. The microbiome is composed of a community of bacteria, viruses and other microorganisms that normally live in the respiratory tract. Viral infection and the use of medical therapies can cause an imbalance in the microbiome, allowing certain bacteria to overgrow or others to decline, leading to selective change. How could this have influenced the results found?

   - Changing environmental conditions: During the pandemic, elderly patients may be confined indoors, such as hospitals or long-term care homes. These environments may have specific characteristics such as inadequate ventilation, high humidity or exposure to cleaning chemicals. These environmental factors may favor the growth of certain bacteria over others, promoting selective changes. How was this fact considered by the authors?

 - Host factors: Elderly patients often have a weakened immune system, which can affect the composition of bacteria present in respiratory sputum. Changes in immunity can create a favorable environment for the growth of certain bacteria, allowing them to selectively proliferate. This discussion must be included in the manuscript.

 - Please, check all references according to the author's instructions.

- Include more details in the figures (error bars) and tables captions.

- The manuscript must be formatted according to the journal's standards.

Minor editing of English language required

Author Response

Reviewer 3 Report (New Reviewer)

However, authors should consider some small suggestions before suggesting their authorization for publication.  

- Although this is mentioned in the article's weaknesses in the discussion section, it is prudent to insist there is the possibility of including the underlying diseases of the studied population.  

- Sputum microbiological cultures are intended to identify causative agents in patients with respiratory symptoms who have come to the hospital. If this is true, it is suggested to include a brief paragraph mentioning the usefulness of knowing the colonization or infection by different isolated microorganisms.  and the clinical results of the cultures are not shown.

- It would be prudent for non-expert readers to briefly describe the parameters used in the Geckler classification since an important part of the analysis results is based on the influence of this classification. 

The text is correctly written and has correct syntax and graphics, with minimal errors. 

Author Response

This manuscript is a resubmission of an earlier submission. The following is a list of the peer review reports and author responses from that submission.

Round 1

Reviewer 1 Report

.

Reviewer 2 Report

All changes were made appropriately.

Reviewer 3 Report

The current work focuses on the Selective change in the bacterial flora in respiratory sputum from elderly patients during the SARS-CoV-2 pandemic. The experimental work appears to have been carried out well. However, a few points deserve attention for further publication. I suggest that it is accepted for publication after the following revisions:

During the SARS-CoV-2 pandemic, the selective alteration of the bacterial flora in the respiratory sputum of elderly patients can be influenced by several factors.

The authors could correlate the results presented in the manuscript with some possible contributions to this alteration, namely:

 - Use of antibiotics: The elderly, especially those with comorbid conditions, may receive antibiotic treatment for respiratory infections secondary to COVID-19. Overuse or inappropriate use of antibiotics can negatively affect the normal bacterial flora of the respiratory tract, allowing overgrowth of certain bacterial strains and reducing microbial diversity.

- Compromised Immune Status: As people age, the immune system can become compromised, making seniors more susceptible to secondary bacterial infections. This can lead to changes in the composition of the bacterial flora present in respiratory sputum.

 - Hospital stay: Older people are more likely to be hospitalized if they contract COVID-19. Prolonged hospitalization can expose patients to different environments and a variety of bacteria present in intensive care units and other hospital sectors. This can lead to a change in the normal bacterial flora of the respiratory tract.

 - Airway hygiene: During the pandemic, there has been significant emphasis on hand hygiene and the use of face masks to prevent the spread of the virus. These measures may have an indirect impact on the bacterial flora of the respiratory tract, reducing exposure to certain pathogens or altering the microbial balance.

 - Lifestyle changes: The pandemic may have caused changes in living patterns, including movement restrictions, social isolation and reduced physical activity. These factors can affect overall health and the immune system, potentially leading to changes in respiratory bacterial flora.

- Please, check all references according to the author's instructions.

 - Include more details in the figures (error bars) and tables captions.

 - The manuscript must be formatted according to the journal's standards.

Minor editing of English language required